# Indoor aeroallergens from American cockroaches and mites initiate atopic march via cutaneous contact in a murine model

**Mey-Fann Lee**[1], **Yu-Wen Chu**[2,3], **Chi-Sheng Wu**[4], **Ming-Hao Lee**[1], **Yi-Hsing Chen**[3,4,5☯], **Nancy M. Wang**[6☯]*

1 Department of Medical Research, Taichung Veterans General Hospital, Taichung, Taiwan, 2 Department of Pharmacy, Taichung Veterans General Hospital, Taichung, Taiwan, 3 School of Medicine, National Yang Ming Chiao Tung University, Taipei, Taiwan, 4 Division of Allergy, Immunology and Rheumatology, Taichung Veterans General Hospital, Taichung, Taiwan, 5 Department of Medicine, National Chung Hsing University, Taichung, Taiwan, 6 Department of Biology, National Changhua University of Education, Changhua, Taiwan

☯ These authors contributed equally to this work.
* nancy@cc.ncue.edu.tw

**Data Availability Statement:** All relevant data are within the manuscript and its Supporting Information files.

## Abstract

The progression of allergic diseases from atopic dermatitis in childhood to other allergic conditions such as asthma in later life is often referred to as the atopic march. In order to study the relationship between cutaneous sensitization by aeroallergen and atopic march, we established a mouse model to test the hypothesis using American cockroaches and house dust mites as the model allergens. Mice were sensitized via skin with native cockroach extract (CraA) or recombinant Per a 2 and Der p 2 proteins without adjuvant. Each mouse was subjected to a total of three 1-week patching sensitizations with a 2-week interval in between each application. The resulting immunological variables in sera, scratching behavior, airway hyperresponsiveness (AHR), and pathology of skin lesions and nasal mucosa were evaluated. In mice, application of CraA, rPer a 2, and rDer p 2 aeroallergens through skin patching induced significantly high levels of both total IgE and specific IgEs. The epicutaneous sensitization after a subsequent allergen challenge showed a significant increase in scratch bouts, AHR, epidermal thickness, and eosinophil counts in the skin compared with the control mice. In addition, stimulation of murine splenocytes with allergens increased higher levels of Th2 cytokines, anti-inflammatory cytokines, and chemokines excretion. Our study provides evidence supporting that epicutaneous sensitization to aeroallergens also led to nasal and airway symptoms comparable to atopic march as described in humans. We hope this new allergy model will be useful in the development of new preventive and therapeutic strategies aimed at stopping the atopic march.

## Introduction

The sequential development of allergic disease manifestations during early childhood is often termed the "atopic march". It describes a typical clinical feature of patients with atopic dermatitis (AD)/skin allergy symptoms in early childhood and subsequently develops allergic rhinitis and asthma/airway allergies [1–4]. There is increasing evidence that other factors influence the subsequent progression of airway allergies, allergic rhinitis, and allergic asthma later in life [5–

**Funding:** *author: Mey-Fann Lee *grants numbers : TCVGH-1107312C (MF Lee) TCVGH-1103802C (MF Lee) and TCVGH-T1107811 (MF Lee) *funder: Taichung Veterans General Hospital *URL: https://bit.ly/3Ny4sp7 *The funders had no role in study design, data collection and analysis, decision to publish, or preparation of the manuscript.

**Competing interests:** The authors declare that they have no competing interests.

8]. As the most common chronic inflammatory skin disease, AD has a lifetime prevalence in US school-age children of up to 18% [9] and affects 6.7% of the general population in Taiwan [10]. Though it more frequently occurs in childhood, it can affect individuals at any age.

Dermal sensitization of food allergens through defective skin has been studied in several murine models. Exposing mice to ovalbumin (OVA) or peanut on abraded skin leads to significant specific IgE responses [11]. Topical application with OVA in mice with loss-of-function mutations in the filaggrin (FLG) gene resulted in cutaneous inflammation and sensitization, as measured by specific IgE to OVA, but not in wild-type mice [12]. Benor *et al.* demonstrated that cutaneous exposure to peanut oil induces systemic and pulmonary peanut hypersensitivity reactions in a mouse model [13]. In a prospective birth cohort study, Lack and colleagues found that low-dose exposure to peanut oil applied to infants with inflamed skin was associated with an increased risk of peanut allergy at an early age [14]. Studies by Brough and colleagues also discovered that peanut allergens in household dust may promote peanut sensitization and eczema severity [15,16]. This impact was even more significant in children with filaggrin loss-of-function mutation [17].

Although this dermal sensitization of food allergen via defected skin can be recapitulated in mice, however, there has been a lack of suitable animal models to study cutaneous sensitization to aeroallergens and subsequent development of AD and airway allergies. We hypothesized that aeroallergens could sensitize the host via cutaneous contact, initiating the atopic march. We established a mouse model to test the hypothesis that atopic march can be triggered by contact with aeroallergens via skin using the leading indoor allergens of American cockroaches and house dust mites as the model allergens.

## Materials and methods

### Preparation of endotoxin removal American cockroach extract and recombinant Per a 2 and Der p 2 proteins

American cockroach extract (CraA) was prepared from *P. americana* with Coca's solution, as described previously [18]. *E. coli*-expressed American cockroach major allergen Per a 2 and house dust mite major allergen Der p 2 recombinant proteins (rPer a 2 and rDer p 2) were purified by rapid affinity column chromatography (Novagen, Darmstadt, Germany) [19]. The CraA and both of recombinant proteins were further purified by Endotoxin Detoxi-Gel (Pierce, Illinois, USA) and sterilized by passing through a 0.22-μm syringe filter (Millipore, Billerica, MA, USA). Finally, the protein concentration was determined by the Coomassie brilliant G-250 protein-dye binding method of Bradford using bovine serum albumin as a standard according to the manufacturer's procedures (Bio-Rad, Hercules, CA, USA).

### Experimental design of allergen-induced atopic march in a murine model

Six-week-old female BALB/c mice from the National Laboratory Animal Center, Taiwan, were used for the experiments. All animal experiments were reviewed and approved by the Institutional Animal Care and Use Committee of Taichung Veterans General Hospital (La-1091738). The scheme of the experimental design is shown in the Fig 1.

### Epicutaneous sensitization and intradermal/intranasal challenge

To induce AD-like skin lesions, 50 μg each of CraA, rPer a 2 or rDer p 2 in 100 μl of phosphate-buffered saline (PBS) were applied on $1cm^2$ gauze pads. The patches were placed on the shaved back skin with a transparent dressing (3M HealthCare) for one week and then removed. Two weeks later, an identical patch was reapplied onto the same skin site. Each

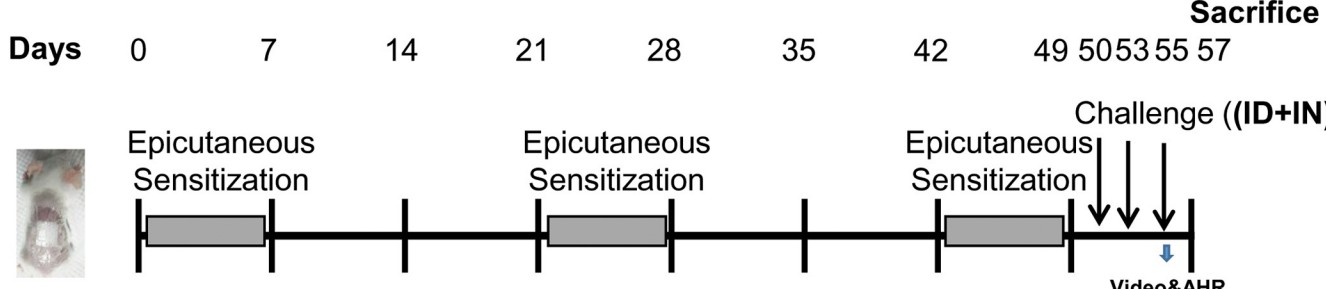

**Fig 1. Experimental design of allergen-induced atopic march in a murine model.** Mice were sensitized via skin with native CraA or *E. coli*-expressed rPer a 2 and rDer p 2 in 100 μl PBS to a sterile patch. The patches were placed for a 1-wk period and then removed. Each mouse had a total of three 1-wk sensitization at 2-wk intervals. All groups were challenged ID and IN on days 50–55 for three consecutive days and sacrificed on day 57.

mouse received a total of three 1-week exposures at 2-week intervals. To check the sensitization response, skin scratching behavior and airway hyper-responsiveness (AHR) were performed on days 55 and 56 after intradermal (ID) and intranasal (IN) challenges, respectively. Control mice received PBS for both patching sensitization and challenge experiments. Serum samples were collected from the submandibular vein bi-weekly and stored at -20˚C until analysis. All mice were sacrificed on day 57, and tissues from skin, nose, and spleen were collected for further study.

### Scratching behavior

After the 3 cycles of allergen patching, the scratching behaviors were videotaped for 1 h immediately following the last ID challenge at the right leg dorsal region on day 55. Each scratching event around the ID site was recorded using video playback [20].

### Measurement of airway hyperresponsiveness (AHR)

On day 56, the AHR of mice was measured using a whole-body Buxco mouse plethysmograph (Buxco, NY, USA) 24 h after the last IN challenge. Mice were placed in the main chamber and challenged with aerosolized methacholine at 0, 25, and 50 mg/mL concentrations generated by a nebulizer (Buxco aerosol distribution system). The degree of bronchoconstriction was measured and averaged for 3 min after each nebulization. Data were expressed as enhanced pause (Penh) by the following equation: Penh = pause × (PEP/PIP). Pause, PEP, and PIP refer to the expiration time, peak expiratory pressure, and peak respiratory pressure, respectively.

### Histological analysis of skin lesions and nasal mucosal tissue

The mice were sacrificed 48 h after the last allergen challenge, and the dorsal patched skins were removed immediately. The excised skin specimens were cut into two pieces. One of the

skin lesions was immediately submerged in RNA stabilization solution (Invitrogen, CA, USA) for RNA extraction. The other piece was fixed in 10% neutral formalin overnight and embedded in paraffin. For nasal mucosal evaluation, the mice were decapitated, and then the heads were fixed in 10% neutral formalin for 72 h. The heads were then washed in running water and placed in Calci-Clear Rapid for decalcification for 72 h. The paraffin blocks of skin and nasal tissues were divided into 4-μm thick sections and stained with hematoxylin and eosin (H&E). All slides were examined under a Hamamatsu NanoZoomer 2.0 HT slide scanner (Hamamatsu, Japan) at 400-fold magnification.

## Analysis of total IgE, allergen-specific IgE, IgG1, and IgG2a antibodies by ELISA

According to the manufacturer's instructions, the total IgE levels were measured using an IgE mouse ELISA kit (Invitrogen). Ninety-six-well MaxiSorp plates (Nunc, Denmark) were coated overnight at 4˚C with a 5 μg/mL monoclonal antibody. All of the following steps were performed at room temperature. After washing twice with PBS/0.05% Tween 20, the plates were blocked with PBS plus 1% BSA for 2 h and then incubated for 2 h with prediluted serum (1:10) or 2-fold serial dilutions of mouse IgE standards. After washing, the detection antibody (1:250) was added and incubated for 2 h. The plates were washed followed by addition of streptavidin-horseradish peroxidase conjugate (1:400) for 30 min. For visualization, tetramethylbenzidine (TMB) substrate was added, and the reaction was stopped with 1 M phosphoric acid. The absorbance was measured at 450 nm using an ELISA reader (TECAN, Austria).

The specific-IgE, -IgG1, and -IgG2a to allergens CraA, rPer a 2, and rDer p 2 from each group were determined by in-house ELISA with the required antibodies purchased from BD Pharmingen (San Jose, Calif). Microtiter plates (Maxisorp, Nunc) were coated with an individual allergen (100 μl from 5 μg/mL in 100 mM $NaHCO_3$, pH9.6). After blocking with 2% BSA, sera were diluted at 1:10 for IgE, 1:1000 for IgG1, and 1:100 for IgG2a and then incubated overnight at 4˚C. For IgE measurement, the plates were incubated with primary biotin-conjugated rat anti-mouse IgE (1:4000) and secondary peroxidase-conjugated streptavidin (1:10000). Detection was performed using TMB and stopped with 1.0 M phosphoric acid. For IgG measurement, the plates were incubated with peroxidase-conjugated rabbit anti-mouse IgG1 (1:10000) or IgG2a (1:10000) and the binding reactions were visualized by adding ABTS solution. The absorbances were then determined using Sunrise Absorbance Reader (TECAN) at 450 and 415 nm.

## Measurement of cytokine production of stimulated splenocytes

Splenocytes were cultured in 24-well flat-bottomed plates at a concentration of $1 \times 10^6$ cells/mL and stimulated with 1 μg/mL of CraA, rPer a 2, or rDer p 2 for each group at 37˚C for three days for RNA isolation. For cytokine assay, cells were stimulated for 4 days and the supernatants were collected and stored at -20˚C. The levels of cytokines IL-1α, IL-1β, IL-2, IL-3, IL-4, IL-5, IL-6, IL-9, IL-10, IL-13, IL-17A, Eotaxin, G-CSF, GM-CSF, IFN-γ, MCP-1, MIP-1α, MIP-1β, RANTES, and TNF-α were evaluated using a Bio-Plex mouse 23-plex panel (Bio-Rad). According to the manufacturer's instructions, protein concentrations were determined based on a standard curve described by the manufacturer-provided protocol.

## RNA preparation and quantitative real-time PCR

Total RNA was extracted from the lungs, skin lesions, and stimulated splenocytes using TRI-ZOL Reagent (Invitrogen, Carlsbad, CA). RNA concentrations and the 260/280 nm absorbance ratio were determined using NanoDrop One ND1000 (Thermo; Waltham, MA, USA).

cDNA was reverse-transcribed using a SuperScript III kit (Invitrogen) according to the manufacturer's protocol using 1 μg total RNA in 20 μL reactions. The quantitative real-time PCR was performed on a StepOnePlus$^{TM}$ system (Applied Biosystems, CA, USA) with primers listed in S1 Table. The gene expression data were expressed as a fold increased between untreated cells and allergen-stimulated cells after normalized with the housekeeping gene β-actin.

## Statistical analysis

Statistical analysis was performed using IBM SPSS software, version 22 (IBM Corporation, Armonk, NY, USA), with appropriate methods. *P*-values less than 0.05 were considered to be statistically significant.

## Results

### Detection of total IgE and allergen-specific IgE, -IgG1, and -IgG2a in sera after epicutaneous allergen exposure

To establish epicutaneous sensitization using purified allergens, we patched mice with cockroach crude extract (CraA), purified rPer a 2, or rDer p 2 recombinant proteins without adding an adjuvant. Quantitative analysis of total IgE was performed to evaluate the sensitized status of the mice. Among the four groups, mice that were patched with CraA, rPer a 2 and rDer p 2 developed significantly higher levels of total IgE at week 8 (Fig 2A) in comparing with the PBS control group.

We further examined allergen-specific serum antibodies before and after skin exposure by in-house ELISA. After patching, CraA- and rPer a 2-specific IgE antibodies were significantly increased at two weeks following sensitization. The rDer p 2-specific IgE level was not significantly elevated until the fourth week (Fig 2B). Only rPer a 2 group showed a marked elevation of allergen-specific IgG1 level in a time-dependent manner. rDer p 2-specific IgG1 was significantly increased at week 8 (Fig 2C). However, our model revealed that Per a 2-specific IgE rose significantly in the early stage and reduced gradually to the basal level at week 8. A similar antibody responsive pattern was also observed in specific IgG2a (Fig 2D).

### Measurement of skin hypersensitivity by scratch counts, cytokine expression, and histopathology after allergen challenge

AD is a chronic skin disorder that is characterized clinically by intense itching and scratching, leading due to allergic inflammation. To examine whether re-exposure to aeroallergens would elicit itching after epicutaneous sensitization, the scratching events were counted from video playback. Fig 3A shows that the mice in the CraA, rPer a 2, and rDer p 2 groups exhibited significantly increased scratching bouts after the allergen challenge compared to the control mice.

We measured cytokines that are known to participate in the progression of AD. IL-13 is one of the major pathogenic cytokines released in response to cutaneous allergens in AD [21]. IL-31 is associated with cutaneous lymphocyte antigen-positive skin-homing T cells and itching in AD patients [22]. Thymic stromal lymphopoietin (TSLP) is a critical pro-inflammatory cytokine in both acute and chronic skin lesions of AD [23]. As shown in Fig 3B, 3C and 3D, mice with rPer a 2-patch exhibited a marked increase in mRNA expressions of IL-13, IL-31, and TSLP in comparing with PBS controls. The rDer p 2-patched mice showed the most significant increase in IL-13 and IL-31 in the skin lesions, while TSLP production was found significantly increased in American cockroach-sensitized groups, CraA and rPer a 2.

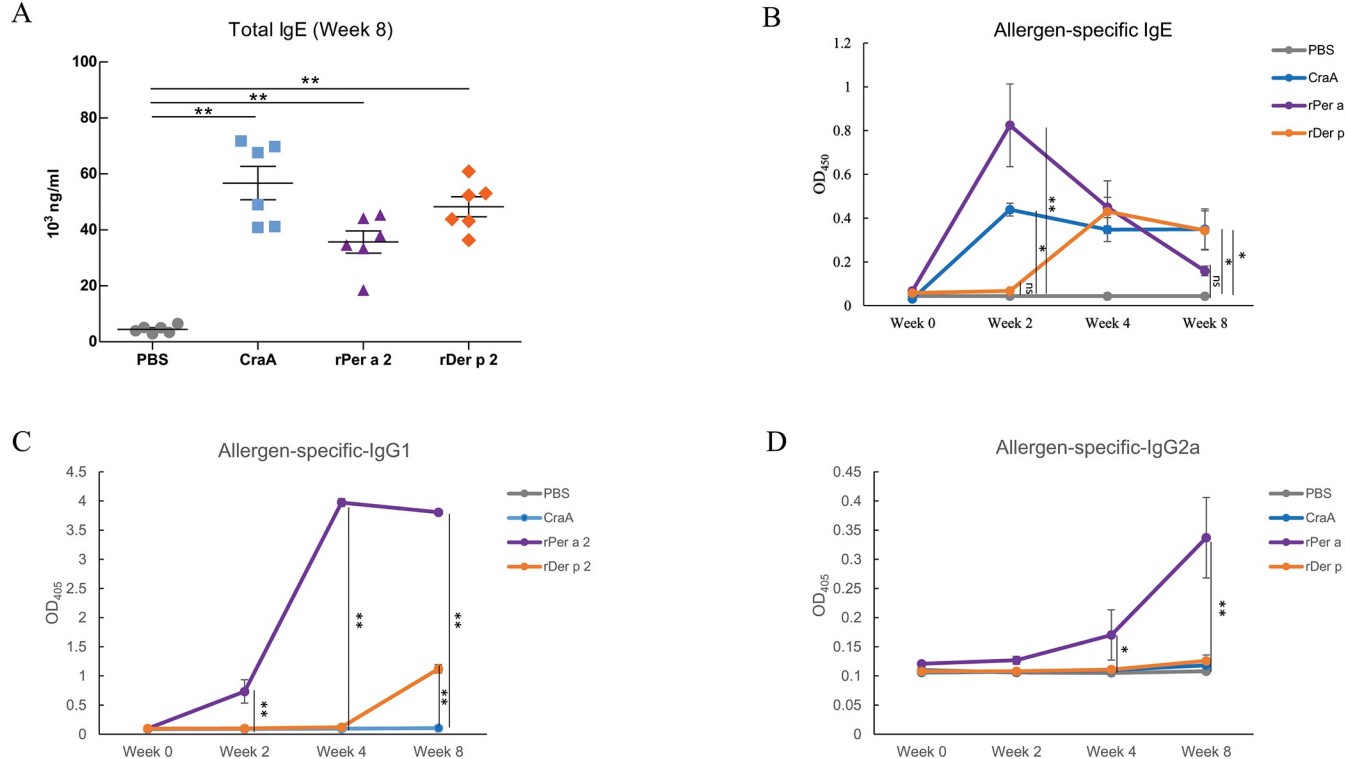

**Fig 2.** Change in serum total IgE (A) and allergen-specific IgE (B), IgG1 (C), and IgG2a (D) antibodies of mice from the four groups as indicated in weeks. Levels of specific IgE and IgGs were expressed as optical density (OD) at 450 nm and 405 nm, respectively. Data were expressed as the mean±SEM of 6 mice from each group. One-way ANOVA with Bonferroni multiple range test: $^*p < 0.05$, $^{**}p < 0.01$, n.s. denoted not statistically significant.

Conversely, skin lesions of AD are characterized by increased skin thickness and dermal infiltration of inflammatory cells. Histological analysis was performed to determine whether aeroallergen-painting sensitization induced immune cell infiltration in AD-like skin lesions. H&E staining showed epidermal hyperplasia and accumulation of inflammatory cells in the dermis of the allergen-patched groups compared to the PBS group (Fig 4A). Repeated patching with either native or recombinant allergens on the skin was able to induce significant epidermal thickness (Fig 4B). Moreover, all of the aeroallergen-treated groups showed a significant increase in total cells, eosinophil, and neutrophil counts compared to the PBS control group (Fig 4C).

## Measurement of allergic nasal inflammation by histopathology after allergen challenge

We then examined whether epicutaneous application of allergens could induce rhinitis-like inflammation in the nose. In the CraA, rPer a 2, and rDer p 2 mice groups after IN challenge, there were mild tissue inflammations with eosinophilia in both the nasal and maxilla turbinates (Fig 5A) but not in the control group.

## Measurement of airway hyper-responsiveness by plethysmograph and pulmonary inflammation by H&E staining

To examine whether or not this epicutaneous exposure of aeroallergens-induced AD mice may lead to allergic asthma, mice underwent IN challenge with specific allergens for each

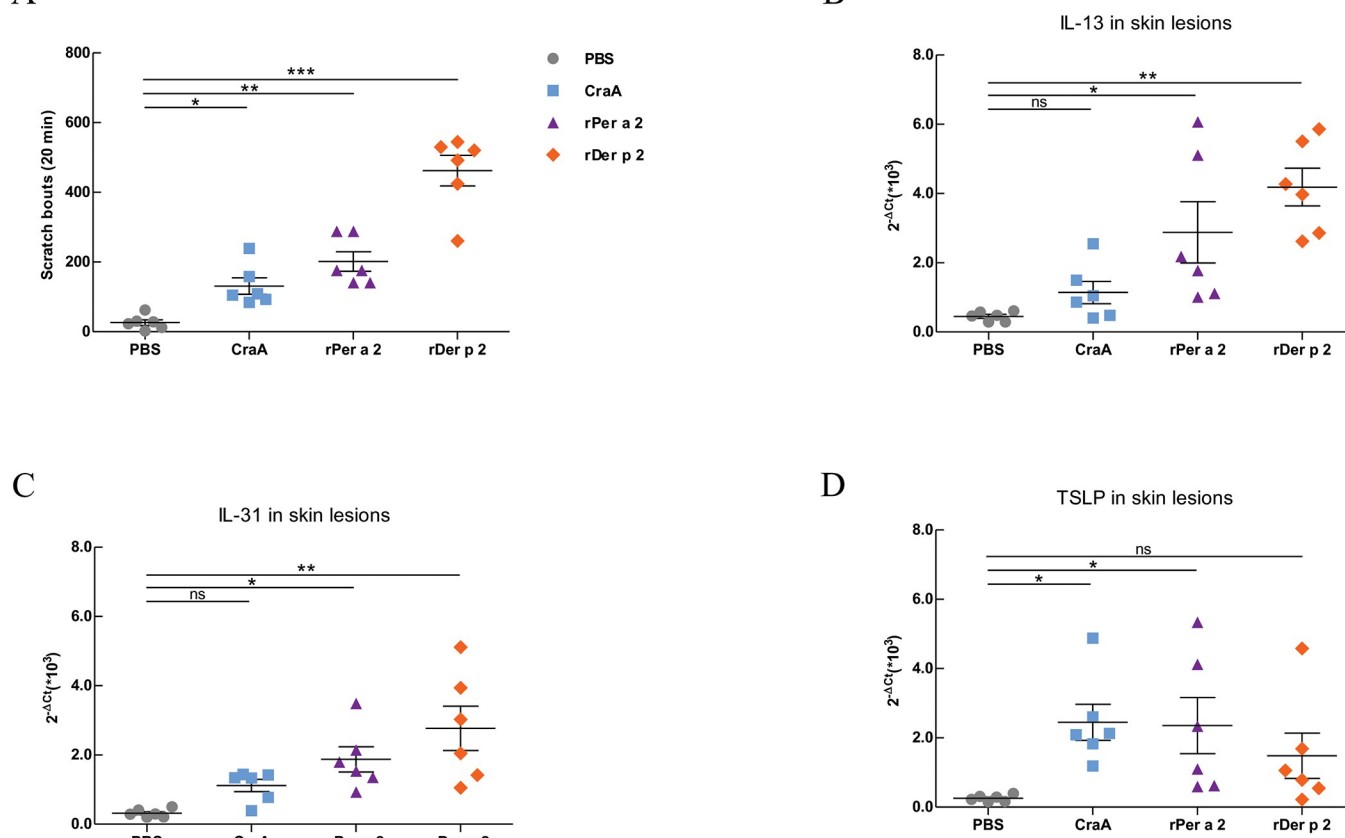

**Fig 3.** (A) Scratching bouts of mice from each group. Expression levels of mRNA for cytokines (B) IL-13, (C) IL-31 and (D) TSLP in skin lesions from the four groups by real-time PCR. The scratching counts were recorded for 20 minutes after induction by ID injection of PBS, CraA, rPer a 2, and rDer p 2, respectively. The statistical significance of differences between PBS and sensitized groups was assessed by the Dunnett's test. Data are mean±SEM of 6 mice. * $p < 0.05$, ** $p < 0.01$, ns denoted not statistically significant.

group. One day after the last challenge, mice were treated with saline aerosol followed by increasing concentrations of methacholine aerosol, and Penh was calculated. In all three sensitized groups, mice showed significantly increased Penh upon methacholine exposure, but not in the PBS group (Fig 5B).

Additionally, histopathologic features of the murine lungs are shown in Fig 5C. The examination of the lung tissues from all three sensitized groups reveals a significant increase in cellular infiltration of total inflammatory cells and eosinophils compared with the PBS group.

## Measurement of allergen-specific systemic inflammation using aeroallergen-stimulated splenocytes

Cytokine expression profiles from splenocytes after stimulation with the individual allergen were analyzed. In the study, cytokines were grouped as follows: Th2-type cytokines/IL-4, IL-5, IL-9, and IL-13; Th1-type cytokines/IFN-γ, TNF-α, IL-1α, IL-1β; anti-inflammatory cytokines/IL-2, IL-10; and chemokines/MCP-1, GM-CSF. The results showed that CraA, Per a 2, and Der p 2 (Fig 6) could induce significantly higher Th2-type cytokines, anti-inflammatory cytokines, and chemokines excretion, but not the Th1-type cytokines. This aeroallergen-induced cytokine profile was also confirmed by real-time PCR as shown in Fig 7A, 7B and 7C.

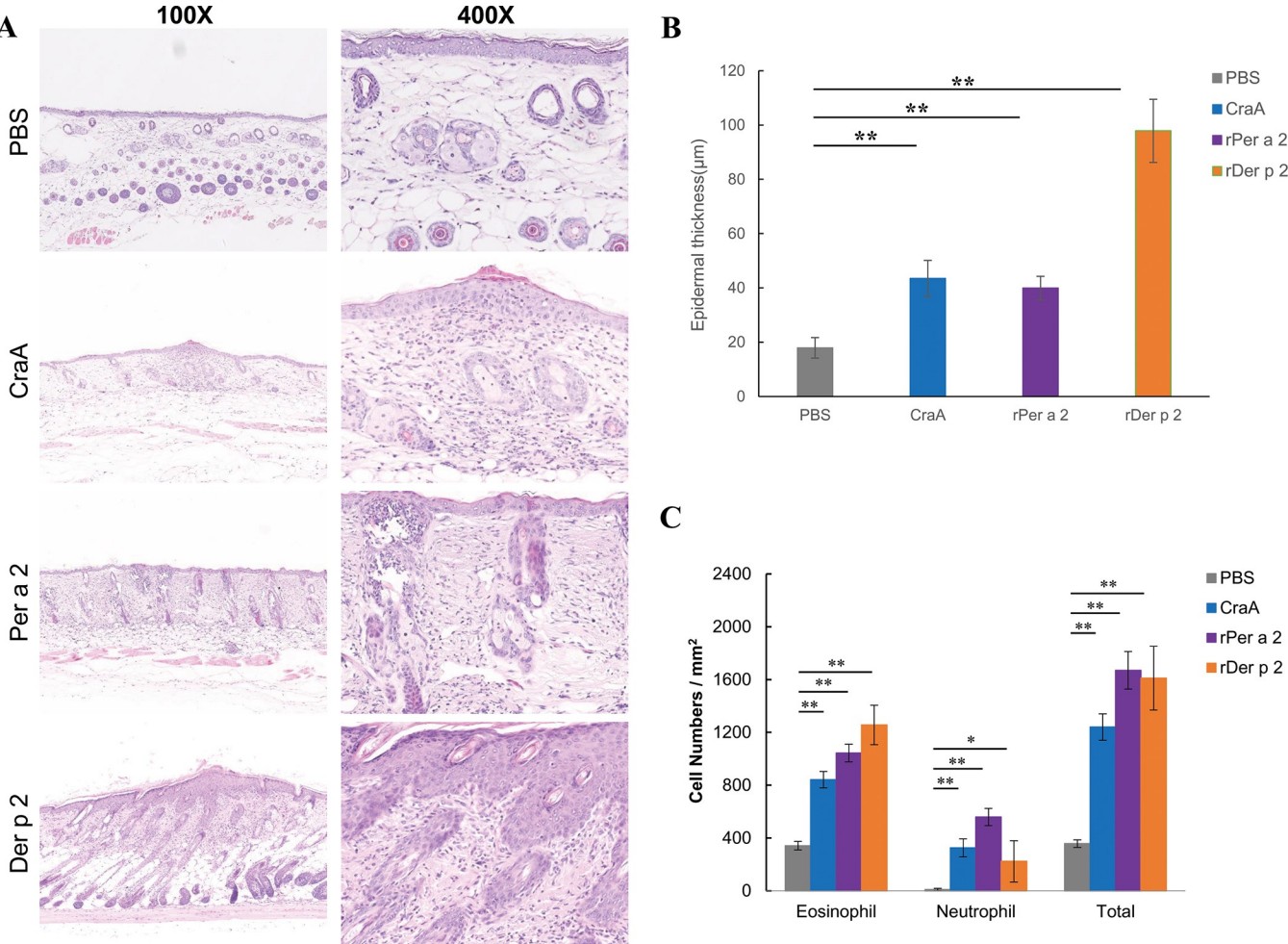

**Fig 4.** (A) H&E staining, (B) epidermal thickness of skin lesions and (C) the cell counts of infiltrating inflammatory cells from each group. Data were expressed as the mean±SD of 5 mice from each group. *$p< 0.05$; **$p<0.01$ by one-way analysis of variance with the Bonferroni multiple range test.

## Discussion

This study used indoor aeroallergens from cockroaches and mites as the model allergens to determine the role of aeroallergens in initiating atopic march via skin. It is well known in respiratory allergy that inhaled sensitization to cockroach and mite allergens plays a critical role in asthma. However, little is known about the initiation and activation of immunity by cockroaches and mite allergens via the skin. In the present study, we found that, in addition to the conventional airway exposure and sensitization, indoor aeroallergens from cockroaches and mites do have the ability to initiate allergic inflammation of the skin as well as nose and lung via cutaneous exposure.

It has been reported that epicutaneous application of house dust mite induced allergic inflammation in non-lesional skin of patients with atopic dermatitis and systemic allergic inflammation in a canine atopic model [24,25]. Our previous studies have shown that different components of cockroach allergens have different allergenicity [26,27]. We found that sensitization to Per a 2 of the American cockroach correlates with greater clinical severity among airway allergic patients in Taiwan compared to other allergen components of cockroaches [26]. Per a 2 is a 42 kDa protein with aspartic protease-like biochemical activity [28]. It is abundant

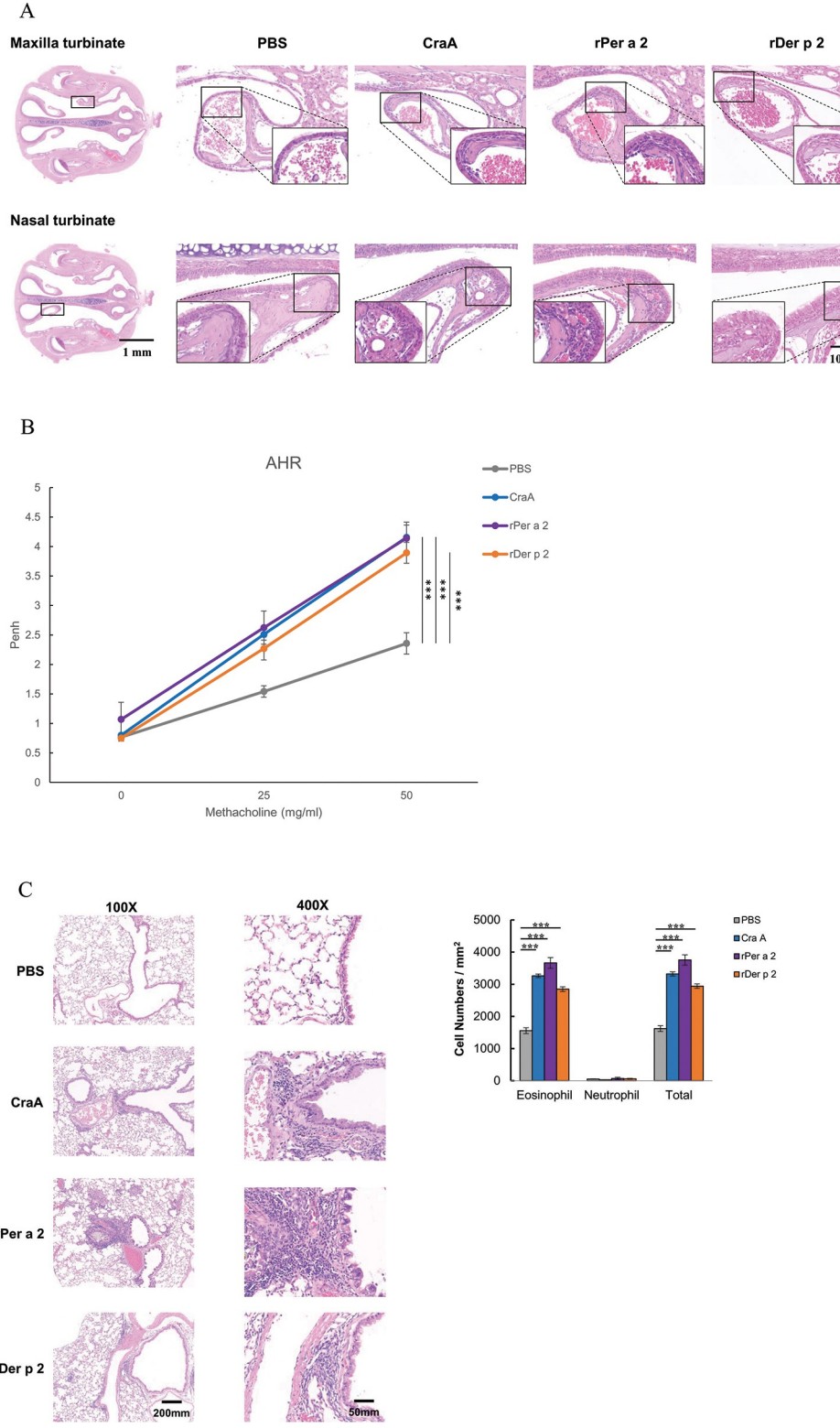

**Fig 5.** (A) Hematoxylin and eosin (H&E) staining of the coronal section of the nasal cavity. (B) Effects of aeroallergens via skin patch on airway hyper-responsiveness (AHR) and (C) lung inflammation by H&E staining in mice. Nasal lateral mucosa of nasal and maxilla turbinate in black squares was shown from each group. Scale bars = 100 μm. Mean enhanced pause (Penh) values were evaluated at 0, 25, and 50 mg/ml of methacholine in the four groups. Data were expressed as the mean±SEM of 5 mice from each group. ***$p < 0.001$ by one-way analysis of variance with the Bonferroni multiple range test.

| Cytokines | PBS control | CraA | rPer a 2 | rDer P 2 |
|-----------|-------------|------|----------|----------|
| IL4 | | | | |
| IL5 | | | | |
| IL9 | | | | |
| IL13 | | | | |
| IFN-g | | | | |
| TNF-a | | | | |
| IL-1a | | | | |
| IL-1b | | | | |
| IL-2 | | | | |
| IL-10 | | | | |
| MCP-1 | | | | |
| GM-CSF | | | | |

ratio

| | |
|---|---|
| | <0.1 |
| | 0.1-0.5 |
| | 0.6-0.9 |
| | 1 |
| | 1.1-3.0 |
| | 3.1-5.0 |
| | 5.1-20 |
| | 21-100 |
| | >100 |

**Fig 6. Aeroallergens-induced cytokine profiles in protein levels of splenocytes from patch-sensitized mice by Multiplex immunoassay.** Mouse splenocytes were cultured under stimulation with individual allergen for 4 days and culture supernatants were analyzed for cytokine release. Color codes in each panel refer to gray for lowest concentrations (<10 pg/mL) and red for the highest protein levels (> 1000 pg/mL) as indicated.

in cockroach feces and is very stable in the household environment. It remains undegraded after a year of decomposition compared to other cockroach allergen components [27,29]. In this study, 50 μg of Per a 2 induced an earlier increase of specific-IgE on week 2 and highest IgG1 level on week 4 than the other same dose tested allergens in initiating allergic inflammation via skin. In general, allergen-specific IgE actives both mast cells and basophils by binding to FcεRI receptor on their surface; however, these cells also express IgG Fc receptor, FcγRIIb, that suppress their IgE-mediated activation. Previous studies showed that sensitization in mice induced both IgE and IgG1 responses [30]. The results echo our previous observations of airway allergy [26].

Allergen exposure is a determinant of atopic march onset. However, it remains controversial whether high or low-dose exposure is more likely to result in allergy (29). It has been reported that a high dose (120 μg/mouse) of cutaneous mite allergen exposure may induce IgG-mediated protection against anaphylaxis, even accompanied by IgE production [31]. However, low-dose exposure to cutaneous cockroaches and mite allergens in this study seemed to induce the allergen-specific IgE and the triad of skin, nose, and airway allergies. Further studies are required to establish whether high-dose exposure can induce more protective Per a 2-specific IgG2a at eight weeks to "neutralize" Per a 2-specific IgE, as shown in Fig 2.

There were some limitations in the current study. First, in order to control the dose of the allergen exposure, we patched the mice for 3 one-week exposures, with two-week interval in between each exposure. In the real world, patients may be exposed to these indoor allergens in a more continuous and persistent pattern. Whether these differences could change the results of allergic inflammation requires further study. Secondly, in our mouse model, the allergic inflammation of the skin, nose, and lung was observed simultaneously after epicutaneous aeroallergen exposure. In the current study, we did not provide a comprehensive depiction of the sequential development of allergic diseases in the atopic march. Our mouse model developed patterns of multimorbidity after skin sensitization with strong allergens which partly supports

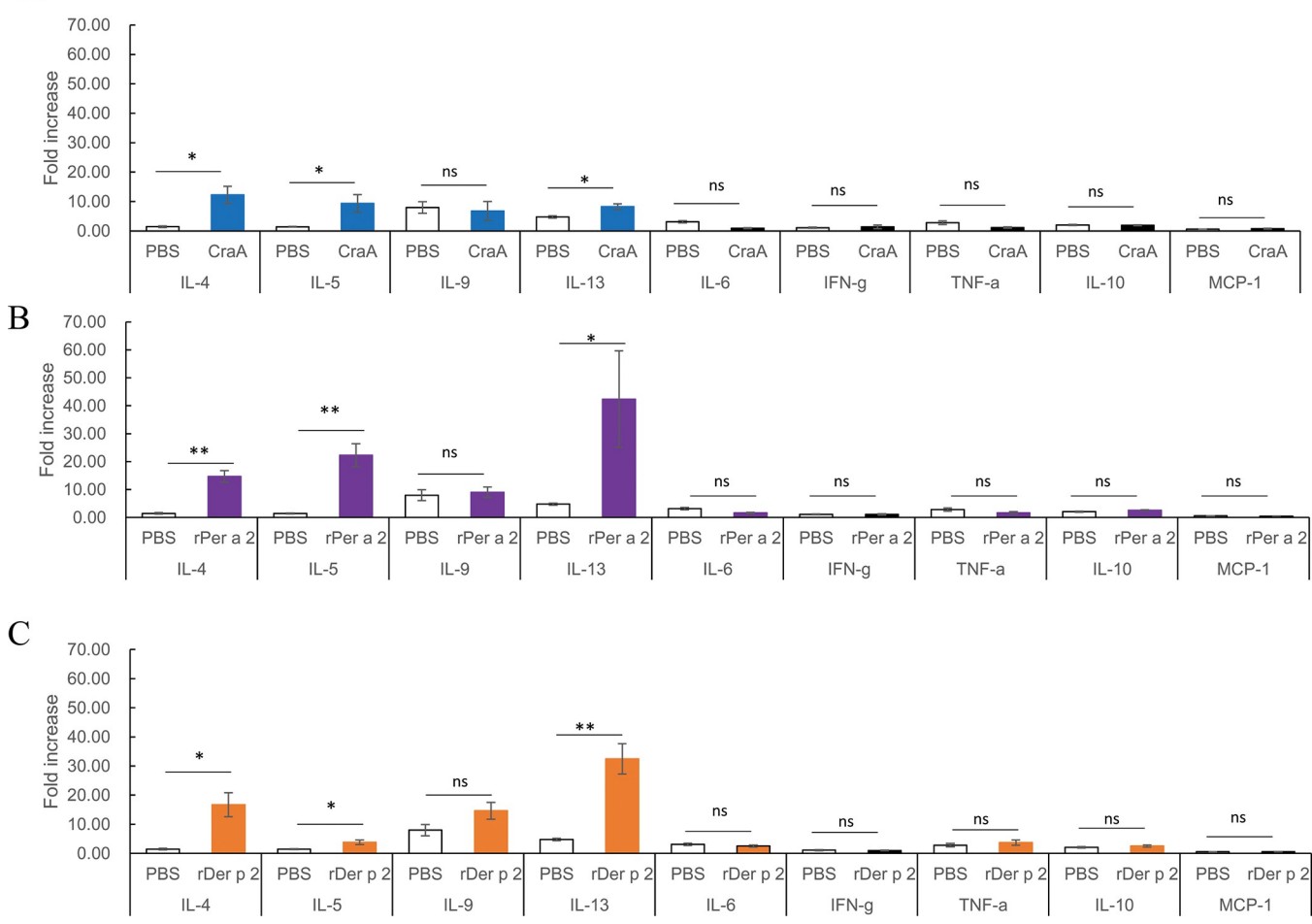

**Fig 7. Aeroallergen-induced cytokine profiles in mRNA expression of splenocytes from patch-sensitized mice by real-time PCR.** (A) different cytokine profiles between control PBS and CraA, (B) PBS vs. rPer a 2, (C) PBS vs. rDer p 2. Mouse splenocytes were cultured under stimulation with individual allergen for 3 days and total RNA were extracted for cytokine expression. Quantitative PCR was performed to determine the mRNA expression of IL4, IL-5, IL-9, IL-13, IFN-γ, TNF-α, IL-6, IL-10 and MCP-1. Fold increase is shown as mean±SEM with respect to untreated cells after correction by the house-keeping gene β-actin. Statistical differences between PBS- and allergen-patching groups were assessed by an independent 2-tailed $t$ test. $^*p<0.05$, $^{**}p<0.01$, $^{***}p<0.001$.

the observation of Haider et al [32]. Thus, in agreement with Yu's discussion [33], the sequence of each allergic disease that occurred in the atopic march progression still remained unclear. However, our *in vivo* evidence did demonstrate that skin exposure to indoor aeroallergens can initiate systemic allergic inflammation in the skin, nose, and lungs. The findings also support the crucial importance of the use of skin emulsions to ameliorate skin barrier dysfunction for preventing or delaying the initiation of atopic march. Finally, in our study, the negative control group received PBS for both sensitization and challenge for comparison. For immune response study, using a non-allergic protein instead of PBS may be a much better alternative.

In conclusion, the current study provides additional evidence supporting the outside-in hypothesis of the development of atopic march. We hope this new allergy model will be useful in the development of new preventive and therapeutic strategies aimed at stopping the atopic march.

## Supporting information

**S1 Table. The sequences of murine gene-specific primers used in real-time PCR.**
(DOCX)

## Author Contributions

**Conceptualization:** Yi-Hsing Chen, Nancy M. Wang.

**Data curation:** Mey-Fann Lee, Yu-Wen Chu, Chi-Sheng Wu, Ming-Hao Lee, Yi-Hsing Chen.

**Formal analysis:** Mey-Fann Lee, Yu-Wen Chu, Chi-Sheng Wu, Ming-Hao Lee, Yi-Hsing Chen.

**Funding acquisition:** Mey-Fann Lee.

**Investigation:** Mey-Fann Lee, Yu-Wen Chu, Chi-Sheng Wu, Ming-Hao Lee, Nancy M. Wang.

**Methodology:** Mey-Fann Lee.

**Supervision:** Mey-Fann Lee, Nancy M. Wang.

**Validation:** Mey-Fann Lee, Nancy M. Wang.

**Writing – original draft:** Mey-Fann Lee, Yi-Hsing Chen.

**Writing – review & editing:** Nancy M. Wang.

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
