## [Decision Letter · Decision Letter 0]

14 Jun 2023

PONE-D-23-13337Indoor aeroallergens from American cockroaches and mites initiate atopic march via cutaneous contact in a murine modelPLOS ONE

Dear Dr. Wang,

Thank you for submitting your manuscript to PLOS ONE. After careful consideration, we feel that it has merit but does not fully meet PLOS ONE’s publication criteria as it currently stands. Therefore, we invite you to submit a revised version of the manuscript that addresses the points raised during the review process.

We look forward to receiving your revised manuscript.

Sincerely,

Nicholas A. Pullen, Ph.D.

Academic Editor

PLOS ONE

Additional Editor Comments:

This is an intriguing study that makes a contribution to the literature on atopic march. You will see that the expert reviewers have provided a favorable position on your study and provided important criticisms. I agree with their assessment. The reviewer points should be addressed in a revision. I look forward to the submission of your revised manuscript.

Reviewers' comments:

Reviewer's Responses to Questions

**Comments to the Author**

1. Is the manuscript technically sound, and do the data support the conclusions?

Reviewer #1: Yes

Reviewer #2: Yes

2. Has the statistical analysis been performed appropriately and rigorously? 

Reviewer #1: Yes

Reviewer #2: Yes

3. Have the authors made all data underlying the findings in their manuscript fully available?

Reviewer #1: Yes

Reviewer #2: Yes

4. Is the manuscript presented in an intelligible fashion and written in standard English?

Reviewer #1: Yes

Reviewer #2: Yes

5. Review Comments to the Author

Reviewer #1: This is an interesting manner to study the atopic march, and the rationale for studying cockroach and HDM allergens is sound. My main question is - How did the authors decide to patch the mice for 3 one week exposures versus a different method? What is the justification for performing the experiment this way?

I would discuss the significance of the IgE and IgG results further.

Line 41 – should be “increase” instead of “increased”

The sentence that starts in line 55 has confusing wording. Perhaps take out “it described a typical clinical feature that” and start the sentence with, “Patients begin with atopic dermatitis…”

Line 183 – “IL-10” is mentioned twice in a row

Line 233 – “in the” should be in the sentence - “known to participate in the progression of AD”.

Line 238 – Should be “shown”, not “showed”

Line 247 – I believe the authors meant to write “epidermal” instead of “epithermal”

Other than what is outlined above, there are some sentences that could be worded better as they are incomplete or lacking.

Reviewer #2: In this study, a mouse model was used to show that epicutaneous sensitization to aeroallergens also led to nasal and airway symptoms comparable to atopic march as described in humans. The approach used was appropriate and the results are interesting. However, there are several points about the hypothesis and the project design that should be noted.

1. It is curious how similar is this model when compared to the situation in humans. As we know, there is a significant time gap in human atopic march. The immediate nasal and airway symptoms triggered by the epicutaneous sensitization in this mouse model is somehow deviated from what we expected.

2. A single dose of CraA, rPer a 2 and rDer p2 has been used for epicutaneous sensitization. How could this dose be justified? Moreover, no dose-dependent effect could be observed if a single dose was used.

3. In this study, PBS was used as the negative control for comparison. However, it is expected that a non-allergic protein should be a better negative control in order to show the sensitization effects of the three agents applied.

6. PLOS authors have the option to publish the peer review history of their article (what does this mean?). If published, this will include your full peer review and any attached files.

Reviewer #1: No

Reviewer #2: No

---

## [Author Response · Author response to Decision Letter 0]

4 Jul 2023

Reviewer: 1

Q1.How did the authors decide to patch the mice for 3 one week exposures versus a different method? What is the justification for performing the experiment this way?

Reply: 

1. For studying the pathogenesis of atopic dermatitis (AD), chemical-induced hypersensitivity of the skin, such as dinitrochlorobenzene (DNCB), is a commonly-used animal model. In order to closely mimic human AD symptoms in response to aeroallergen stimuli, we adopted the sensitization protocol for AD mice from Spergel’s study.* Spergel’s group developed this murine AD model by 3 one-week allergen applications epicutaneously (EC) and was able to induce asthma using methacholine challenge. A similar sensitization protocol using the same 3 one-week application with house dust mite was also developed by Kawakami’s group.* However, Kawakami’s protocol has a shorter intervention time in between each application since the animal used in this study was a dermatitis-prone mice. 

2. Previous observations, from both Spergel’s and Kawakami’s groups, described EC sensitized mice that exhibit local allergic dermatitis and asthma after allergen challenging. In our study, specific allergens were used for both sensitization and challenge. Our results demonstrated the disease progression from systemic AD to allergic rhinitis and asthma on this animal model. Furthermore, allergic symptoms and the course of the disease displayed clinical features similar to those of patients with atopic march.

＊References: 

Spergel JM, Mizoguchi E, Brewer JP, Martin TR, Bhan AK, Geha RS. Epicutaneous sensitization with protein antigen induces localized allergic dermatitis and hyperresponsiveness to methacholine after single exposure to aerosolized antigen in mice. J Clin Invest. 1998 Apr 15;101(8):1614-22. doi: 10.1172/JCI1647. PMID: 9541491; PMCID: PMC508742.

Kawakami Y, Yumoto K, Kawakami T. An improved mouse model of atopic dermatitis and suppression of skin lesions by an inhibitor of Tec family kinases. Allergol Int. 2007 Dec;56(4):403-9. doi: 10.2332/allergolint.O-07-486. Epub 2007 Sep 1. PMID: 17713360.

Q2. I would discuss the significance of the IgE and IgG results further.

Reply: 

In general, allergen-specific IgE plays a primary role in immediate hypersensitivity reactions. Both mast cells and basophils express high affinity IgE receptor, FcϵRI, on their surface; however, these cells also express IgG Fc receptor, FcγRIIb, that suppress their IgE-mediated activation. Previous studies showed that sensitization in mice induced both IgE and IgG1 responses (Oshiba, 1996). Immunohistological studies revealed Th2-dominant responses developed in these sensitized mice, as occurs in humans with allergic disorders (Chen, 1998; van Neerven, 1996). The significance of the IgE and IgG results is addressed in discussion section (Line 307).

＊References: 

Oshiba A, Hamelmann E, Takeda K, Bradley KL, Loader JE, Larsen GL, Gelfand EW. Passive transfer of immediate hypersensitivity and airway hyperresponsiveness by allergen-specific immunoglobulin (Ig) E and IgG1 in mice. J Clin Invest. 1996 Mar 15;97(6):1398-408. doi: 10.1172/JCI118560. PMID: 8617871; PMCID: PMC507198.

Chen YL, Simons FE, Peng Z. A mouse model of mosquito allergy for study of antigen-specific IgE and IgG subclass responses, lymphocyte proliferation, and IL-4 and IFN-gamma production. Int Arch Allergy Immunol. 1998 Aug;116(4):269-77. doi: 10.1159/000023955. PMID: 9693276.

van Neerven RJ, Ebner C, Yssel H, Kapsenberg ML, Lamb JR. T-cell responses to allergens: epitope-specificity and clinical relevance. Immunol Today. 1996 Nov;17(11):526-32. doi: 10.1016/0167-5699(96)10058-x. PMID: 8961630.

Q3. Please make corrections or provide response to the following: Line 41-should be “increase” instead of “increased”. The sentence that starts in Line 55 has confusing wording. Line 183-“Il-10” is mentioned twice in a row. Line 233-“in the” should be in the sentence-“known to participate in the progression of AD”. Line238- Should be “shown”, not “showed”. Line 247- I believe the authors meant to write “epidermal” instead of “epithermal”.

Reply: 

Thank you for pointing out the inappropriate grammatical errors. We apologize for these typo errors and the words on lines 41, 183, 233, and 247 have been corrected. The sentence in Line55 has been rephrased.

Reviewer: 2

Q1. It is curious how similar is this model when compared to the situation in humans. As we know, there is a significant time gap in human atopic march. The immediate nasal and airway symptoms triggered by the epicutaneous sensitization in this mouse model is somehow deviated from what we expected.

Reply: 

Indeed, mice have a much shorter lifespan compared to humans. In order to minimize the age gap relation between mice and humans and increase their clinical relevance, we correlated of age in different life stages in mice and humans based on Dutta’s paper. * The approximate age of mice in this study at the mentioned time points is: week 6: 12yr-old in human; week 8: 20 yr-old in human; week 15: 30 yr-old in human. Thus, clinical features, histopathology, and biochemistry assessments were analyzed from puberty to adulthood in mice.

＊References: 

Dutta S, Sengupta P. Men and mice: Relating their ages. Life Sci. 2016 May 1;152:244-8. doi: 10.1016/j.lfs.2015.10.025. Epub 2015 Oct 24. PMID: 26596563.

Q2. A single dose of CraA, rPer a 2 and rDer p2 has been used for epicutaneous sensitization. How could this dose be justified: Moreover, no dose-dependent effect could be observed if a single dose was used.

Reply: 

The sensitization dose for each recombinant protein/crude extract have been studied in other papers. Since it is not the primary focus of this study, we followed the sensitization protocol and dose of the above mentioned to establish animal model. 

Q3. In this study, PBS was used as the negative control for comparison. However, it is expected that a non-allergic protein should be a better negative control in order to show the sensitization effects of the three agents applied.

Reply: 

Thank you for your comments. We agree that using a non-allergic protein is a better negative control than PBS. We have incorporated this as an additional study limitation addressed in discussion section (Line 338).

---

## [Editor Report · Decision Letter 1]

12 Jul 2023

Indoor aeroallergens from American cockroaches and mites initiate atopic march via cutaneous contact in a murine model

PONE-D-23-13337R1

Dear Dr. Wang,

We’re pleased to inform you that your manuscript has been judged scientifically suitable for publication and will be formally accepted for publication once it meets all outstanding technical requirements.

Sincerely,

Nicholas A. Pullen, Ph.D.

Academic Editor

PLOS ONE

Additional Editor Comments (optional):

I have reviewed your changes in response to reviewer minor concerns, and I am happy to endorse the manuscript for publication.
---

## [Editor Report · Acceptance letter]

19 Jul 2023

PONE-D-23-13337R1 

Indoor aeroallergens from American cockroaches and mites initiate atopic march via cutaneous contact in a murine model 

Dear Dr. Wang:

I'm pleased to inform you that your manuscript has been deemed suitable for publication in PLOS ONE. Congratulations! Your manuscript is now with our production department. 

Kind regards, 

on behalf of

Dr. Nicholas A. Pullen 

Academic Editor

PLOS ONE